# How Will Sense of Values and Preference Change during Art Appreciation? †

**Akinori Abe [1,*,‡]** , **Kotaro Fukushima [2,‡] and Reina Kawada [3]**

1   Faculty of Letters, Chiba University, 1-33 Yayoicho, Inageku, Chiba 263-8522, Japan
2   Graduate School of Humanities and Studies on Public Affairs, Chiba University, 1-33 Yayoicho, Inageku, Chiba 263-8522, Japan; ayga4147@chiba-u.jp
3   Number, Inc., Tokyo 150-0022, Japan; x.xzr07@gmail.com
*   Correspondence: ave@ultimaVI.arc.net.my or ave@chiba-u.jp
†   This paper is an extended version of our presentation in the International Conference on Data Mining Workshops (ICDMW), Beijing, China, 8–11 November 2019.
‡   These authors contributed equally to this work.

**Abstract:** We have conducted several experiments where various types of information offering strategies were performed. We obtained interesting phenomena from the results. The participants seemed to be able to gradually understand the artwork by offering information of the artwork. Of course, for an abstract art, the information of the artworks functions better understanding of artworks. Even for a representational painting, the quality and quantity of understanding was gradually changing. Thus, the information of art sometimes influences the art appreciation. In this paper, we will discuss how the value and preference of art will change according to offered information? In addition, we will discuss determining which factor (information) will change the viewers' value and preference of art in the art appreciation. For that, we conducted two experiments, where information of the artwork was offered randomly (each person may obtain different information for the artswork). Additionally, for all the artworks, the information was offered in the same manner (all persons will obtain the same information for the artworks). The information involved title, painting materials, techniques, production year, name of artist, price, background, and theme of the artworks.

**Keywords:** art appreciation; sense of value; preference; curation

## 1. Introduction

For the art appreciation in museums, certain information will usually be provided as a caption. Visitors usually read the description to help his/her understanding. Museums usually prepare such descriptions for general visitors. The problem for reading these captions is that visitors will not see the artworks after reading the descriptions to understand the art works. However, several museums have recently removed or hidden such descriptions in captions and titles. For instance, the exhibition "Bacon and Caravaggio" held in Museo e Galleria Borghese, Roma, Italy during 2 October 2009 and 24 January 2010, were the case. No information was offered as captions.

Leder et al. proposed a stage model for aesthetic processing, which combines aspects of understanding and cognitive mastering with affective and emotional processing [1]. Subsequently, in [2], they pointed out that "[a]ccording to the model, aesthetic processing of an artwork involves a number of processing stages, which might somehow proceed sequentially and therefore allow the formulation of hypotheses concerning time sensitive processing of art. After initially classifying a stimulus as an artwork, features such as colour, shape, contrast, etc. are analyzed in the perceptual

processing stage". Afterwards, they discussed that " [i]f understanding and grasping the meaning is essential, as proposed in the model, then information which helps to interpret the image must affect aesthetic processing. Here we present a study in which we investigate how verbal information affects cognitive and affective components in the processing of abstract and representational artworks". Thus, if suitable information is given to us, it is possible to interpret the image suitably in the aesthetic processing.

For the above problems, Tadaki and Abe conducted several experiments [3,4]. The experiments showed several interesting results. One of them was that a title sometimes influences visitor thinking. In the experiment, a title was intently hidden. Uncovering the title and after reading the title, one of the participants created the new story according to the title. Actually, it was an abstract painting and it was rather difficult to understand what was painted on the artwork. This phenomena can be regarded as the effect of information. By reading the title, information included in the title was obtained by the viewer to understand the artwork and create the story.

In addition, Abe and Tadaki conducted another experiment, where information of the art was gradually offered [5]. In the experiment, the participants seemed to be able to gradually understand the artwork. Even a representational painting, the level of understanding was gradually changing. The detailed results of the experiment will be shown in the following section. The observation from the result was that the information of art sometimes influences the art appreciation.

In [6], Abe discussed how the value of artworks is created. He used several paintings (on PC) (Figure 1), including Jean Dubuffet's painting and paintings regarded as Art Brut.

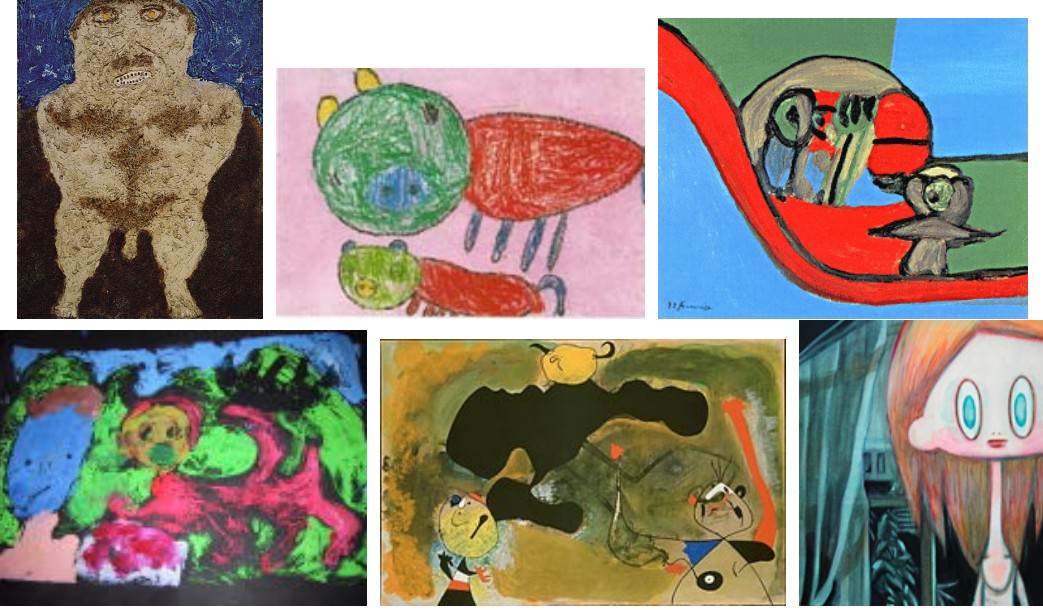

**Figure 1.** Several paintings (downloaded from various sites more than 10 years ago).

Abe mentioned that, by the discovery of Jean Dubuffet, paintings regarded as Art Brut were given value as fine arts. He also used these paintings in his lecture to see the students' response. They could not determine the value of artworks, since the artworks seemed to be something painted by very young children. In addition, since he continued the similar questions, they were rather suspicious. Actually, some were painted by children in kinder gardens. However, the first painting was painted by a very famous painter, Jean Dubuffet. Jean Dubuffet is also very famous for discovering and giving a value to art works called Art Brut. Thus, by connoisseurs of paintings, several artworks happen to have values as good artworks. Additionally, such artworks will come to be liked by an ordinary person.

Subsequently, our question is how will the value and preference of art be changed according to offered information factors? People may value the artworks according to their sense of values

or information surrounding the artworks. For instance, Willem de Kooning's works are not easy to understand. Without certain knowledge, for instance, its price, ordinal peoples will not value his artwork. Because, for novice, it looks like a graffiti or a child's drawing. Similar to the case of Art Brut, after the recognition by the well known person, the value of artworks is sometimes changed. Thus, once the artworks are sold in an auction at big prices, the artworks might be valued even by ordinal peoples (non-experts). For the price of artworks, for instance, Findlay mentioned that "[l]ike currency, the commercial value of art is based on collective intentionality. There is no intrinsic, objective value (no more than that of a hundred-dollar bill). Human stipulation and declaration create and sustain the commercial value". [7] Actually, only a price is not related to aesthetics. For the price, it is sometimes regarded as a rather vulgar matter [8]. Hook quoted Robert Hughes' phrase "[t]he price of a work of art is an index of pure, irrational desire, and nothing is more manipulable than desire". However, such a price can be an influential factor in art appreciation and determination of the sense of values.

In this paper, we will discuss which factors will change the viewers' sense of value and preference of art.

## 2. Gradual Information Offering Strategy

In the previous paper [9], we reviewed two types of curations. One exhibition had no information as captions. The other exhibition had a lot of information. For the exhibition "Information or Inspiration? Japanese aesthetics to enjoy with left side and right side of the brain", we discussed information on captions that "[i]t was very interesting curation. However for the white way, we felt too much information. we felt it was better to provide information in a suitable level. Since the exhibition is for general visitors, it would be the best way to show as many information as possible". For the exhibition, "Bacon and Caravaggio", Coliva addressed that "[t]his exhibition proposes a juxtaposition of Bacon and Caravaggio. It intends to offer visitors an opportunity for an aesthetic experience rather than an educational one... [10]". It is important to offer viewers an opportunity for an aesthetic experience. Subsequently, our question is how much information will be necessary for viewers.

Abe and Tadaki [5] conducted an experiment. In this experiment, our question was how the description of caption would influence or help the visitor's thinking. In the previous experiment [3,4], only one pattern of caption added to the artwork was offered to one participant. In this experiment, all patterns of caption were offered to all participants. That is, all of the participants took several experiments for the artwork.

We used a painting that is shown in Figure 2. The experiment was conducted as a question on a paper. Since the artist will not be famous in Japan, we think, first, no information was shown, and then the information for the painting (title) was shown. At last, information of the artist (name, birth place and history) was shown. When participants finished answering a question, the next paper with the similar question was given. We prepared a question with three levels of information, as follows (prior to the following questions, questions about the user's profile (How often do you go to (art) museums? etc.) were conducted. However in this paper, the answers to the questions were not in consideration);

1.  Please imagine and write the story occurring in the painting.
2.  The title of this painting is "New Day", please imagine and write the story occurring in the painting. If something is different from what you imagined in the previous question, please write them.
3.  The painter who drew this painting is a female painter living in Beograd in Serbia. Her name is Ivana Živić. She was born in 1979 in Sarajevo. She drew this painting in 2018. Please imagine and write the story occurring in the painting. If something are different from what you imagined in the previous question, please write them.

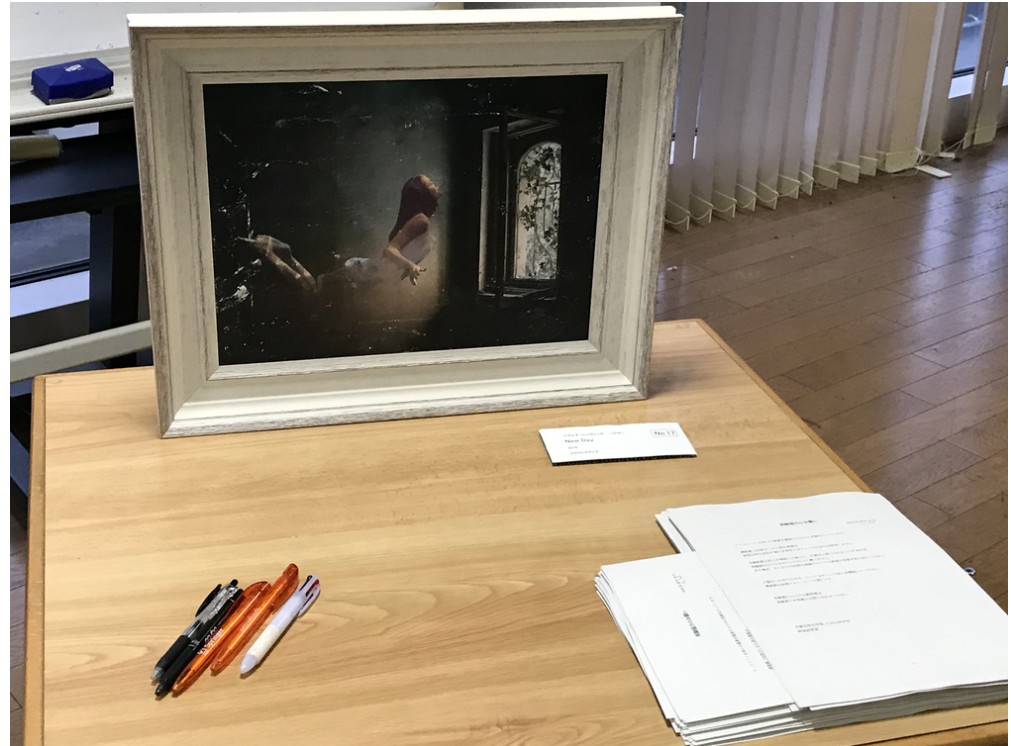

**Figure 2.** Experiment.

That is, all of the participants answered similar questions three times. Information regarding the painting would increase as they proceeded the experiment. They took the same experiments three times. They were given time as long as they liked. Time of one session (three answers) was from 30 min. to 45 min. Their common understandings were:

1. Phrases, such as "the girl has been confined" and "she is floating" were appeared.

    Many of the participants mentioned that she was in water. In fact, for the last three years, she has been working on series of paintings called "Rooms of Water". Unlike the other painting, though this painting does not explicitly paint water, they felt a girl in water. Perhaps the painting shows a certain floating feeling.

2. Phrases, such as "go to the new world" and "go out", were appeared.
3. Since, in the artist's introduction, such a description as "Ivana Živić was born in 1979 in Sarajevo (BIH)" was included, they tended to think the story in the painting from the context of the Yugoslavia conflict. Phrases, such as "escape", appeared.

The participants seemed to be able to gradually understand the artwork. Especially after obtaining the profile of the painter, they could add stories according to the profile (birth place). From the experiment, we could observe that information functions for the better and fruitful understanding. Since this is a representational painting, it will be easy to create a story. However after obtaining the background of the painting, they added or changed their stories according to the information. This type of story creation and understanding will occur, even for an abstract painting.

## 3. Experiment 1

In the previous experiment [5], the participants seem to be able to gradually understand the story or background of artwork according to the quantity of information. at first, we did not show information. Subsequently, we showed the title of the artwork. After that, we showed the profile of the

painter and the production year of the the artwork. Since we did not have an objective to determine the participants' sense of values, we did not ask about the sense of values and preference for the artworks.

Our questions are, as follows:

Is the art appreciation influenced by information, such as price, etc.?
Which factor will make viewer change the value of art?

In this case, "value" means preciousness or preference.
Does the influence differ according to viewers' aesthetic value–intention scale?

We conducted the following experiment in order to solve the above problem. Our hypothesis was that "those who highly have aesthetic sense of values tend to put a high value on an internal elements such as a background of the artwork, techniques of the artwork, and a motif of the artwork rather than external elements such as a price of the artwork and a public reputation".

Since the process and result of this experiment were reported in [9], we briefly review the process and analyze the result.

### 3.1. The Experiment

The experiment was conducted in a room in Chiba University on 24 January 2019. The room could be used as an art gallery (Figure 3). The participants were 20 adults including university students and aged person, and their age was from 18 to 64 years old. The number of females was 13 and that of males was seven. Their naked visions or corrected visions were normal. We used 16 artworks that were created in from 1512 to 2018 as stimuli. Six of them were classical paintings and others were contemporary paintings (six of human beings, three of abstract painting, three of still life, three of animals, and one of landscape). All of the artworks were numbered but were displayed without captions and labels.

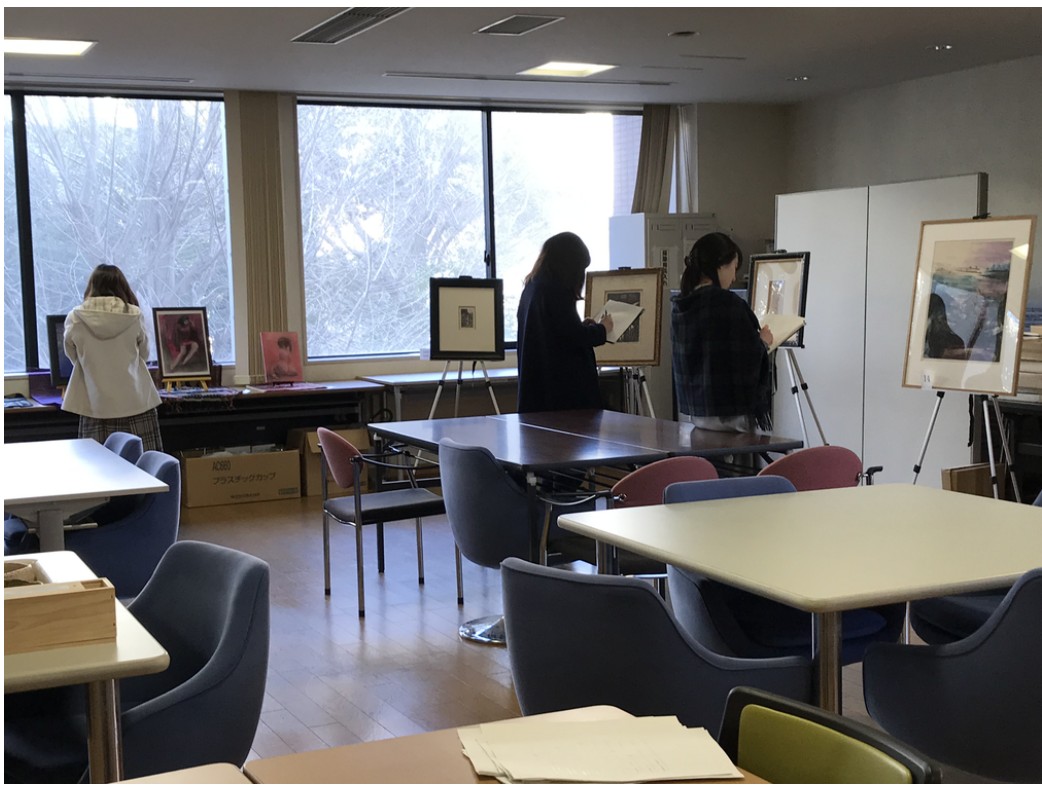

**Figure 3.** Experiment in the room.

The experiment was conducted in the following order: first, the participants answered the questions, including educational and enjoyable experiences of art. From the answer, we obtained the participants' familiarity of art. In addition, the participants answered additional questions (these questions were created with referring to the questions in [11]). Those questions were analyzed for determining the participants' aesthetic sense of values.

Subsequently, the participants evaluated the preference and the value of artworks. In addition, they wrote their impression on the artworks in the worksheet. Actually, they were gradually and randomly shown the information of the artwork. Evaluations were conducted by using the SD method with the score of 1 to 10 being used. For instance, the score 1 for dislike and the score 10 for like. The score 1 for value is low and the score 10 for value is high. During the experiment, the participants freely selected artworks to write their impressions.

*3.2. Analysis of the Result*

We collected answers what participants wrote during the experiment. Several interesting results were obtained.

The total score of the aesthetic sense of values evaluation scales and the case where a participant obtains a higher score or average score in the art-experience questionnaire were correlated.

For the most of participants, the influence of the information disclosure on the evaluation of artworks was not observed.

When no information is shown, those who highly have an aesthetic sense of values put relatively high value on the artworks. However, as an information disclosure proceeds, for some of those who have a highly aesthetic sense of values, a score of the preference becomes relatively high. However, for those who have a lower aesthetic sense of values, a score of the preference is the same.

As for the painting of an animal, many participants scored the preference relatively high. On the other hand, as for the painting of human beings, many of the participants scored the value relatively high. There are a few differences in the scores between abstract paintings and landscape paintings.

For the lower priced artworks, after the price is shown the score of preference decreases. This result might show that without price, the participant likes the artwork, but, after knowing the price, he/she becomes to be disappointed or dislike the artwork. His/her preference might be influenced by the price. However, no significant result was observed as for a value.

We could not observe that, as for the preference, the positive and negative correlation seems to change according to the aesthetic sense of values. That is, even when the information of price of the artwork is shown, the preference of the artwork is hard to change according to the price. However, as for the value, a negative correlation could be observed, regardless of the aesthetic sense of values. That is, after the information of the price is shown, it is observed that the price is higher, the score of value goes down more significantly. Perhaps, since most of participants were university students, this phenomenon was observed. They will not know the actual price of artworks.

From the results, for a certain part, we can observe the influence of the information on the value and preference of artworks.

## 4. Experiment 2

In the previous experiment, each of the participants were gradually and randomly shown information of the artwork. Accordingly, they might have had different information on the same artworks. It will be difficult to determine which type of information influence their sense of values or preference.

*4.1. The Objective of This Experiment*

In this experiment, we would like to determine which factor (information) will change the viewers' sense of values and preference of art in art appreciation. Therefore, we decided to offer the same information on the same artworks to each participant.

### 4.2. Participants

Participants were 27 adults, including university students and aged person and their age were from 21 to 65 years old. The number of females was six and that of males was 21. Their naked visions or corrected visions were normal. All of the participants were asked to answer about their art education background. These were the same as the questions in the previous experiment [9].

### 4.3. Stimuli

The experiment was conducted in a room in Chiba University on 21 December 2019. The experimental environment was the same as the previous experiments' environment. The room could be used as an art gallery. We used 24 artworks created in from 1896 to 2019 as stimuli. We selected artworks by not-so-famous-in-Japan (we thought) painters in Japan. All of the artworks were numbered, but they were displayed without captions and labels. Signatures were hidden for some artworks. This was because we would like to remove preconceptions.

### 4.4. Method

All of the participants answered the questions on the worksheet during appreciating the artwork. The questions were:

- About this artwork, please write freely what you think, what you feel, and a story in the artwork.
- Do you like this artwork?
- Do you think this artwork is valuable?
- (after information provided) If you have something different after you obtain the information, please write down.

In [3], we observed that frequent museum visitors tended to answer that information about the artist's life, historical background, and theme of the artwork were helpful. 44% of non-frequent museum visitors who wrote a comment about contribution answered that information about techniques used on the artwork was helpful. Accordingly, we prepared information regarding techniques and artist's life. In addition, both positive and negative information are prepared.

Our hypothesis is that:

For the positive information, value and preference of the artwork will increase. Additionally, for the negative information, value and preference of will decrease.

For instance, the following were prepared:

- The artist's father is very famous (positive).
- The artist used a very special technique, for instance, a wood engraving and an Encaustic painting (positive).
- The artist ruined himself by gambling (negative).
- The artist's comment on his/her artworks (neutral?).

We will show the a part of statements we showed to the participants as information.

For the artwork 1, we provided a statement, such that "her father is a very famous painter Georges Rouault". In addition, we prepared Georges Chabot's statement that " [a]t this she succeeds, thanks to her gift for harmonious aesthetics...".

**positive information: the artist's father was famous.**

**positive information: good comment by a critic.**

For the artwork 2, we provided a statement, such that "this painter's name is Syohachi Kimura. In 1912, he took part in the formation of the Hyuzan-kai with Ryusei Kishida. In 1918, he won the Takayama Tyogyu prize. ... This artwork was painted in his closing years".

**positive information: the artist was famous person in a group.**

**positive information: he won the famous prize.**

**positive information: this is almost the last work.**

For the artwork 3, we provided the curator's statement, such that " she expresses the life activity and Reincarnation by Japanese paper and fire. She controls freely to changes the shape of a Japanese paper, which has an expression, such as a human skin. In addition, by using fire to burn the paper, she represents light and shadow on her artwork. ... A moth always appears in her artworks. ... She uses it as an icon of life and death to discipline her expression".

**positive information: the technique is splendid.**

**positive information: theme of the artwork.**

**rather negative point: a moth.**

For the artwork 4, we provided the curator's statement, such that " what is interesting in his work is that from a certain period, for his creation theme, world class theme as the Socialist bloc began to be selected. In short, he topicalized the political, social, and cultural situation of Cuba which is the socialist state".

**positive information: the artist's creation attitude.**

**negative point: rather cheep and outdated creation.**

For the artwork 5, we provided the statement, such that "his father is the famous potter Tou Morita (the copy master of Koryo tea bowl). He learnt ceramic art from his father and created splendid works. But we heard that the artist ruined himself by gambling".

**positive information: the artist's father was famous.**

**positive information: the artist had good education.**

**negative information: the artist ruined himself by gambling.**

For the artwork 7, we provided the statement, such that "he is now a student in the Graduate School, Tama Art University. ... He is a fun of Makoto Aida and Keiichiro Hirano. He is creating his artworks under the theme of unusuality".

**negative information?: the artist is a student.**

For the artwork 9, we provided a statement, such that " he worshiped Rembrandt Harmenszoon van Rijn so much to write a letter to Rembrandt and write a reply with an encouraging from Rembrandt by himself. For his portrait, he adopted the composition similar to Rembrandt's. By the advanced technique in the oil painting, he always drew radical and dense images which came into his mind".

**positive information: the artist was influenced by the famous artist.**

**positive information: the artist has the advanced technique in the oil painting.**

**negative information: the artist is rather paranoia.**

For the artwork 10, we provided the statement, such that "This painting is on the mook book entitled "The world of beautiful women paintings by Japanese painters (Tatsumi publication)". In addition, we provided the artist's comment about the painting. We also add an information that this artwork was the last finished one for her personal exhibition".

**positive information: the artwork is appeared in a mook book.**

**positive information?: this artwork was the last finished one for her personal exhibition.**

**negative information?: the artist's comment is rather philosophical.**

For the artwork 12, we provided the statement, such that "this small artwork was painted by the technique Encaustic. Encaustic involves using heated beeswax to which colored pigments are added. The liquid or paste is then applied to a surface. The wax encaustic painting technique was used from the 1st Century AD".

**positive information: the artist used the very special and rarely used technique.**

**rather negative point: this work is rather difficult to understand.**

For the artwork 15, we provided the artist's statement, such that " when I draw flowers, I would like to draw not only the flower as it is but also an atmosphere emerged by the flower. ... Previously, I was said by viewer that "In your work, I feel the floweriness and sparkled image of a flower more than the actual flower". Which is the best compliment".

**positive information: the artist's view of drawing.**

**positive information: the viewer's good comment.**

For the artwork 17, we provided information that the artist's artwork was selected as a series of illustrations for the story serially appeared in the Asahi. In addition, her exhibition information as url. Additionally, we also provided her attitude that she consistently draw a painting as the subject of the scenery. She create her artworks based on actual scenery that she draws on the sketchbook. She pursuits of painting a landscape that reflects the time surrounding us and changing phenomenon.

**positive information: the artist's artwork was selected as a series of illustrations for the story serially appeared in the Asahi.**

**positive information: the artist's attitude in creation.**

For the artwork 18, we provided the statement, such that "he is one of the representative artists in Cuba. The perfection of his artworks which is backed by his vast quantities of learning and practice is outstanding in the Cuban artists many of who are known as excellent artists".

**positive information: the artist is very famous in his country.**

**positive information: the artist's technique is splendid.**

For the artwork 19, we provided the statement, such that "this woodcut print is produced by using the technique wood engraving". In addition, we prepared the artist's statement such that "when I create a minute print, I activate my nerves in fingers and foot. This is only because I was very nervous".

**positive information: the artist used the very special and rarely used technique.**

**positive information: artist's strict attitude in creation.**

For the artwork 20, we provided the artist's statement, such that "I draw an abstraction and a scenery with an inspiration from the nature". In addition, we prepared artist's attitude in creation such that he thinks the harmony with the nature and peace of mind are important. In addition, we provided the artist's comment such that " although the canvas is small (F1), I feel the colour advocates powerfully and gently. The image of the painting is Winter. However, we can see the coming spring ahead". We also provided his career that in 2008, he studied a painting by himself.

**positive information: the artist attitude in creation.**

**rather negative information: the artist studied a painting by himself (no academism).**

In addition, related books and magazines were on the other table and the related artworks (e.g., an etching by Georges Rouault) were also exhibited.

*4.5. Procedure*

The experiment followed the following procedure:

1.  We obtained the participants' familiarity of art by questions.

    The question set includes was the same as that used in the previous experiment.

2.  We determined the participants' aesthetic sense of values.

    This procedure was the same as that in the previous experiment. However, in this paper, we do not use this information.

3.  The participants evaluated the preference and the value of artworks and wrote their impressions on the artworks without any information (even artist's name).

    The evaluations were conducted using the SD method with the score of 1 to 5 was used. For instance, the score 1 is for dislike, the score 5 is for like, and the score 3 is for neutral. Score 1 is for low value and the score 5 is for high value.

4.  The participants evaluated the preference and the value of artworks with certain information. In addition, if their evaluations were different from those without any information, we asked them to write the reason and the other impressions.

    The evaluations were also conducted by using the SD method with the score of from 1 to 5.

The experiment was conducted in a room in Chiba University. The participants wrote answers on the answer sheet (paper) during art appreciation. The order of the art appreciation was not fixed.

*4.6. Result and Analyses*

We collected answers what participants wrote during the experiment. Several interesting results were obtained. In this paper, we will only analyze the score part. The free descriptions will be analyzed and discussed in the next paper.

Figure 4 shows score (preference and value) differences before and after information was shown. Each graph involves several stacked bars. A central bar means that no difference occurred after the information was shown. The right side bar means that the participants gave more value or preference score after the information was shown. The left side bar means that the participants gave less value or preference score after the information was shown. Almost half of the participants did not change their score after the information was shown. However, for the value, several participants changed their score after the information was shown. For instance, for artworks 1, 2, 3, 4, 5, 9, 10, 12, 15, 18, 19, and 20, the participants tended to increase their score after the information was shown. For artworks 3, 7, 10, and 17, the participants tended to decrease their score after the information was shown. For the artworks 3 and 10, both of the tendencies (increase and decrease) are observed.

For the preference, several participants changed their score a little after the information was shown. For instance, for artworks 1, 3, 5, 7, 12, 15, 16, 19, and 20, the participants tended to increase their score after the information was shown. For artworks 7, 11, 16, and 17, the participants tended to decrease their score after the information was shown. For the artworks 7 and 16, both of the tendencies (increase and decrease) are observed.

In the followings, we will analyze the score of individual artworks.

First, many of the participants did not change their mind after reading information. However, some of the participants change their sense of values of the artworks. For instance, Isabelle Rouault is not famous in Japan. Additionally, we hid her signature during the experiment. In fact, her painting is rather abstract, but we could recognize some fruits. Many of the participants added the better score after reading information. For the artwork 5, similar phenomenon can be observed. Although we

added negative information, this information did not influence the participants mind a lot. In Japan, artists who take narcotic drugs will be excluded from the society. Even their works are excluded from the society. However, in this case, such a case did not occur. If the splendid technique is mentioned in the information, the participants tend to add score in value.

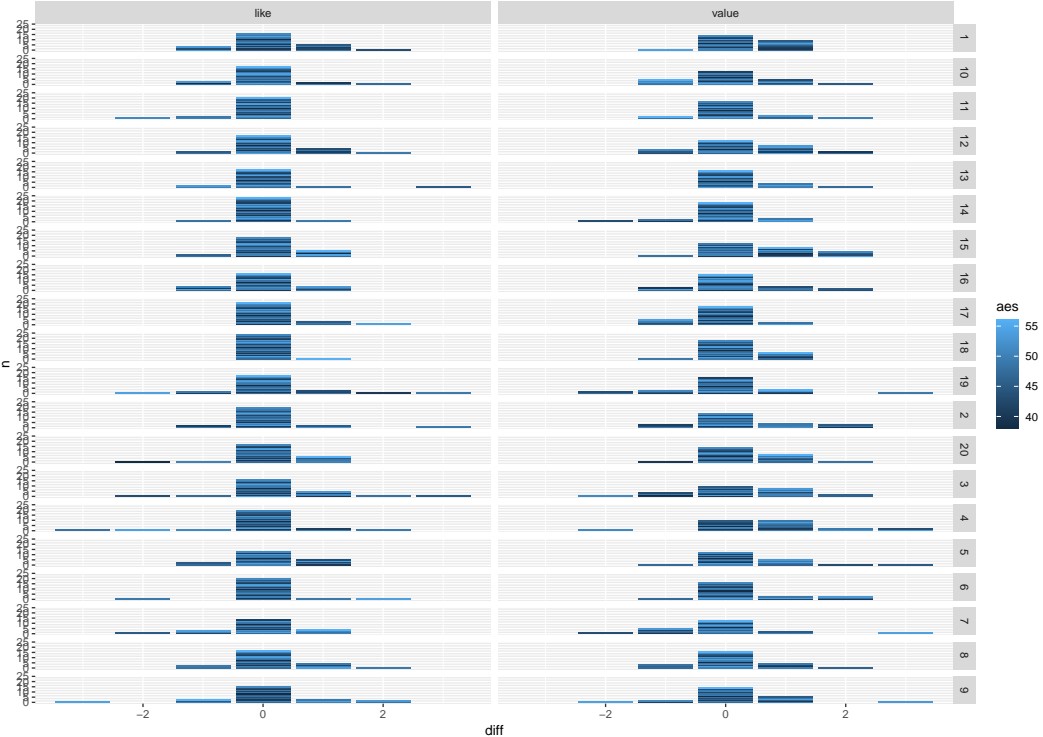

**Figure 4.** Score (preference and value) differences before and after information is shown.

For the preference of the artworks, not so big changes are observed. We think it is natural situation.

The artwork 15 is a Japanese painting of flowers, but a rather abstract painting. After reading the information, some added the better score.

Thus, for the sense of values, if we are provided the respectable information, such as his/her father, leading person and splendid technique, we think the artwork more valuable than our thought. For the preference, it is rather difficult to change the mind. However, if the artist's lovable comments are obtained, we tend to like the artworks better.

For the sense of values, our hypothesis is supported with positive information. However, for negative information, such a tendency was not observed. For the preference, our hypothesis is not supported.

Actually, if we consider the comments from the participants, we can conduct better analysis. For instance, our hypothesis will be analyzed by the participants' comment much better. This analysis will be conducted in the next paper [12].

## 5. Conclusions

This paper discussed the hypothesis that whether those who have high aesthetic value tend to put a high value on an internal elements, such as a background of the artwork, techniques of the artwork, and a motif of the artwork, rather than external elements, such as a price of the artwork and a public reputation. In addition, this paper discussed which factor (information) will change the viewers' sense of value and preference of art in art appreciation. For that, we conducted two experiments by preparing an environment, like an art gallery.

We are still analyzing the result, but obtained interesting knowledge. That is as follows:

Those in the higher scored group appreciated artworks with predicting the information of the artworks. This is because they have knowledge on artworks in order to predict the information. Accordingly, it is considered that those in the lower scored group passively appreciate artworks, but those in the higher scored group actively appreciate artworks when considering the meaning.

For the sense of value, if we are provided the respectable information, such as his/her father, leading person, and splendid technique, we think the artwork more valuable than thought. For preference, it is rather difficult to change the mind. However, if the artist's lovable comments are obtained, then we tend to like the artworks better. A similar situation will occur in the other application, such as a data jacket.

Additionally, such a suggestion that, for the more fruitful art appreciation, it will be necessary to obtain the knowledge of artworks, to experience art appreciation many times, and to be interested in artworks was obtained.

This is the first trial to reveal the relationship between an aesthetic value and knowledge of art. Several interesting results could be obtained.

In this paper, the participants' comments are analyzed as they are. We did not use a tool, such as text mining tool. We separately analyzed the obtained data (scare and comments). It is necessary to analyze the score and comments together. Thus, in future study, we will analyze the obtained data more. In addition, we would like to conduct an additional experiment in order to obtain more knowledge on the problem.

**Author Contributions:** Conceptualization, R.K. and A.A.; methodology, R.K. and A.A.; software, K.F.; validation, A.A. and K.F.; formal analysis, K.F. and A.A.; investigation, R.K.; resources, A.A.; data curation, R.K., K.F. and A.A.; writing—original draft preparation, A.A.; writing—review and editing, A.A.; visualization, K.F.; supervision, A.A. All authors have read and agreed to the published version of the manuscript.

**Funding:** This research received no external funding.

**Conflicts of Interest:** The authors declare no conflict of interest.

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
