# Peer review of "How Will Sense of Values and Preference Change during Art Appreciation?†"

_information, doi:10.3390/info11060328_

Round 1

Reviewer 1 Report

The quality of the English is extremely low, to the point that the paper is not always easy to understand.

Section 1 on exhibitions can be shortened considerably, as much of the detail provided is not very relevant to the paper.

For Experiment 1, I would like to have a little bit more detail about the design of the experiment, in particular about the specific nature of the information revealed.

Conversely, for Experiment 2, more detail than necessary is provided.  In particular, the list of art works and information provided on pages 9-10 can be reduced, I think, to a handful of art works, say 4 or 5.  This will give the reader a sufficient idea of the nature of the information provided to the experimental subjects.

Author Response

Thank you for your meaningful review.

>> The quality of the English is extremely low, to the point that the paper is not always easy to understand.

We corrected English some parts.
Some answers from Japanese participants are quite difficult to translate.

>> Section 1 on exhibitions can be shortened considerably, as much of the detail provided is not very relevant to the paper.

In the section, we addressed the importance and influence of the information at
the captions. However, we removed them. And added some statements on the beginning of the next chapter.

>> For Experiment 1, I would like to have a little bit more detail about the design of the experiment, in particular about the specific nature of the information revealed.

Since it was written in the previous paper, we removed a certain part of the description of the Experiment 1.
But according to your advice I add the details.

>> Conversely, for Experiment 2, more detail than necessary is provided. In particular, the list of art works and information provided on pages 9-10 can be reduced, I think, to a handful of art works, say 4 or 5. This will give the reader a sufficient idea of the nature of the information provided to the experimental subjects.

The list of information of the artwork is important. Because by the list,
it can be determined which type of information can influence viewers mind.
We did not put list for the all artworks.

Reviewer 2 Report

Very original and convincing paper! 

Author Response

Thank you for your kind review.

Reviewer 3 Report

The paper's composition is coherent; the structure is logical and meets the goal of the paper. The title "How will sense of values and preference change during art appreciation?" put well the paper's objective; it is clear and expresses the issue being assessed very well. The abstract is formulated adequately along with the true picture of the paper; all required components are mentioned there. All the tools and methods the author uses are reasonable and well described and adequately fit the problem being assessed to give the reliable results. Conclusions are related to the results presented before reflecting the assessed issue at a professional level. All the figures are complete and understandable. Author uses enough figures featuring a great deal of data being processed from experiments and questionnaire surveys hence adding a higher added value to the paper. I found the paper well-written and cohesive. Author appears to be a professional, very well oriented and involved in the issue. The length of the paper is adequate to the significance of the topic. No other modification is necessary. Paper is clear and understandable and the overall level of language being used is very good. This manuscript has a logical layout and its content is suitable for international scientific journal and that's why I recommend it to be published in Information journal.

Author Response

Thank you for your kind review.